# Effect of mode of healthcare delivery on stress and intention to quit among physicians in Canada during the COVID-19 pandemic

**Hossam Ali-Hassan**[1]☯*, **Shauna Clayton**[2]☯, **Safoura Zangiabadi**[2]☯

**1** Department of International Studies, Glendon Campus, York University, Toronto, Canada, **2** School of Kinesiology and Health Sciences, Keele Campus, York University, Toronto, Canada

☯ These authors contributed equally to this work.
* hossama@yorku.ca

**Data Availability Statement:** Data and survey details from Statistics Canada can be found at: https://www23.statcan.gc.ca/imdb/p2SV.pl?

## Abstract

The COVID-19 pandemic prompted adaptations to the delivery of healthcare services across Canada. In response to associated health risks and physical distancing protocols, some physicians adopted telemedicine procedures into their practice where possible. The present study aimed to investigate the impact that mode of healthcare delivery had on physicians' intention to quit their jobs due to stress, burnout, or mental health. The study utilized data collected by Statistics Canada from the Health Care Workers' Experience (SHCWEP) survey during the COVID-19 pandemic. The sample included 2,198 participants, weighted to represent 93,952 Canadian physicians aged 18 and above. Modes of healthcare delivery were categorized as either in-person, online, or blended. A multivariable logistic regression analysis was performed to examine the relationship between mode of healthcare delivery and intention to quit due to stress, burnout, or mental health, after adjusting for sociodemographic, job-, and health-related factors. Intention to quit within the next two years due to stress, burnout, or mental health was reported by 7.5% of physicians. Compared to the in-person modality, online or blended healthcare delivery was associated with decreased the odds of intention to quit (OR = 0.67, 95% CI: 0.63–0.72 and OR = 0.66, 95% CI: 0.58–0.75, respectively). The present study sheds light on factors associated with medical frontline worker well-being and retention, factors which can subsequently impact the quality of patient care. Future considerations regarding healthcare policy should incorporate strategies that protect and enhance physicians' mental health into its framework to mitigate future risks.

## Introduction

The Coronavirus 2019 (COVID-19) pandemic sparked an unprecedented transformation in Canadian healthcare infrastructure. Federal public health recommendations for reducing the spread of infections prompted provincial governing agencies to enact physical distancing protocols. Following these recommendations, many healthcare systems across Canada adjusted their clinical policies and procedures to conform to public health regulations. These

Function=getSurvey&SDDS=5362 https://www150.statcan.gc.ca/n1/en/catalogue/13250006.

**Funding:** The authors received no specific funding for this work.

**Competing interests:** The authors have declared that no competing interests exist.

adjustments were established to allocate medical resources efficiently and protect staff and patients from contracting infections [1]. Aside from urgent cases where physical distancing was not medically appropriate, these changes brought to light the implementation of telemedical appointments, including telephone, video conferencing, and text messaging services [2]. Telemedicine enabled a multidisciplinary approach that facilitated the triage, assessment, consultation, diagnosis, and remote monitoring of patients by medical professionals [3]. Data collected by Statistics Canada indicated that 24% of healthcare practitioners opted to provide virtual care to their patients during the pandemic [4]. These services varied between telephonic (87%), video (47%), and message communication in the form of email, text, or instant messaging (26%). Among medical frontline workers, some physicians reported challenges in adjusting to the digital workload and concerns regarding adequacy and the quality of patient care [5]. However, other research suggests that some physicians were more optimistic about telemedicine. For example, they found it easier to decline patient requests they deemed unnecessary [6]. To date, little is known about the impact modes of delivery had on stress, burnout, and mental health and the intention to quit the workforce among physicians.

COVID-19 interim protocols became an essential adaptation to medical practice but also posed some problems for medical professionals. For instance, the impromptu deployment of telemedical procedures entailed little planning or preparation, given the rapidly changing pandemic circumstances. Thus, COVID-19 and subsequent risk mitigation tactics, although beneficial in some ways, added complexity to an already challenging job. For instance, a pre-pandemic survey indicated that physicians were at risk of burnout and strain on mental health, with 34% scoring positive for depression and 30% reporting high levels of overall burnout [7]. More recent studies suggest that COVID-19 may have exacerbated these symptoms, perpetuating high levels of stress and exhaustion [8]. Healthcare workers faced a myriad of stressors, including exposure to the virus, lack of personal protection equipment, social isolation, and increased workload [9]. Studies have also underscored the impact of pandemic-induced heavy workloads on medical professionals' mental and physical well-being. In a 2021 survey by the Canadian Medical Association (CMA) assessing physician health and wellness during the pandemic, results showed that 76% of respondents felt "somewhat" or "very" fatigued, with 69% reporting that their fatigue increased over time since the start of COVID-19 [10]. Furthermore, beyond the workload, fear of contracting COVID-19 has been negatively linked to job satisfaction and intention to exit the workforce among healthcare professionals. A study out of Poland assessing 97 physicians at the start of the pandemic showed high levels of stress and fear of COVID-19 to be associated with negative job satisfaction [9]. Many studies have also examined the impact of these factors on influencing the decision to quit their job [e.g.,11]. Additionally, the intention to quit has been linked to stress and burnout in the absence of adaptive coping strategies, such as positive reframing and emotional support among physicians [12]. Although many studies have assessed the direct mental health impact that COVID-19 had on physicians and other medical professionals' job satisfaction and well-being, studies on more indirect pathways from working to quitting are lacking. Accordingly, such an example can be seen in the alterations to the medical delivery framework that may challenge personal and professional functioning.

## Aim of the study

The aim of the study was to examine the association between the mode of healthcare delivery and physicians' intention to quit their jobs due to stress, burnout or mental health concerns during the pandemic in Canada.

## Materials and methods

This study was a secondary data analysis of the Health Care Workers' Experiences During the Pandemic (SHCWEP) survey conducted by Statistics Canada [13]. The SHCWEP study was cross-sectional in nature and was conducted during the period spanning September to November 2021. In Canada, during that period, 80% of eligible Canadians were fully vaccinated. However, the Delta variant caused a resurgence in cases, leading to the reintroduction of some restrictions, especially for unvaccinated individuals in indoor settings [14]. The study's main aim was to gather comprehensive insights into the impact of the COVID-19 pandemic on healthcare professionals within Canada. The study encompassed healthcare workers aged 18 and above, employed within healthcare settings since the onset of the COVID-19 pandemic, and residing within the ten provinces of Canada. Among the estimated population of 22,293 eligible participants, a total of 12,246 individuals completed the survey, resulting in a response rate of 54.9%. Participation was voluntary, and data was collected through electronic questionnaires and computer-assisted telephone interviews. The present research study, however, specifically focused on physicians who had participated in the SHCWEP survey.

The dependent variable of the study assessed physicians' intention to resign from their positions due to stress, burnout or mental health concerns. The data related to this variable was obtained by asking, "How long are you planning to stay in your current job?". Participants who expressed an intention to leave their current job within two years were further queried about the reasons behind their decision, utilizing the question, "What are the reasons that you might consider leaving or changing your job?" participants whose responses indicated an intent to quit their jobs specifically due to job stress or burnout or mental health concerns were categorized as "Yes," while those intending to quit for other reasons or expressing no desire to quit were categorized as "No".

The primary independent variable for this study was the mode of healthcare service delivery during the COVID-19 pandemic, as ascertained through the question, "Since March 2020, how did you provide healthcare services to patients or clients at your primary job location?" this variable was categorized into three broad categories: in-person, online (encompassing video meetings, email, text, or instant messaging), and blended (combining both in-person and online modes). Furthermore, this study accounted for several covariates, including job-related factors (years worked in the job, number of job locations, type of healthcare job, receipt of formal training in Infection Prevention and Control (IPC), sociodemographic factors (gender, age, province of residence, household income, immigration status), and health-related factors (perceived general health).

The analysis encompassed descriptive statistics regarding the intention and reasons for physicians contemplating quitting their jobs within two years. Chi-square tests and bivariate logistic regression models were employed to examine the relationship between each of the job-related, sociodemographic, and health-related factors and physicians' intentions to quit their jobs due to stress, burnout or mental health concerns. Additionally, a multivariable logistic regression model was conducted, wherein the outcome variable was the intention to quit one's job due to mental health concerns, and the primary independent variable was the mode of healthcare service delivery, controlling for all relevant job-, sociodemographic-, and health-related variables. The results were reported in terms of Adjusted Odds Ratios (ORs) and 95% Confidence Intervals for the final model. Population weights were applied to each estimate, and bootstrapping was employed to adjust for the complex sampling methodology utilized in the survey. All statistical analyses were conducted using the Statistical Package for the Social Sciences (SPSS, version 28.0), with a predefined significance level of alpha set at 0.05.

### Ethics statement

This study did not require ethics review by the York University Ethics Board as SHCWEP public use microdata files produced by Statistics Canada are publicly accessible [4] and appropriately protected by law via the Data Liberation Initiative [15].

## Results

The total number of physicians analyzed in this study was 2,198, weighted to represent 93,952 Canadian physicians. Table 1 indicates the descriptive statistics of physicians' intention to quit their jobs within two years and the corresponding reasons. Among respondents, 20.8% expressed the intention to quit their job within two years, with 7.5% attributing their decision to leave the profession due to stress, burnout or mental health concerns. Table 2 presents the descriptive statistics of physicians' characteristics, along with the bivariate and multivariate associations between mode of healthcare delivery, job-, socio-demographic-, and health-related factors with intention to quit their job due to mental health reasons. Of the participants, 50.9% were males, and 31.5% were 55 years and older. Also, 38.5% of respondents resided in Ontario, and more than 75% reported an income of $\geq 150,000$. Among participants, 32.9% delivered healthcare services in person, 7.6% provided online services, and 59.4% offered blended healthcare services to patients.

The multivariate logistic regression results showed a significant association between the mode of healthcare service delivery and the intention to leave the profession due to stress, burnout, or mental health concerns (Table 2).

When compared to physicians who provided an in-person mode of healthcare delivery to patients, those who provided online or blended healthcare services exhibited lower odds of intending to quit their jobs because of stress, burnout or mental health concerns (OR = 0.67, 95% CI: 0.63–0.72; OR = 0.66, 95% CI: 0.58–0.75, respectively). Among job-related factors, intention to leave the profession was significantly associated with the number of years worked and the various locations worked. Participants with less than ten years of work experience had a 1.5 times higher likelihood of intending to quit their jobs than those with twenty years or more of work experience. Moreover, participants who worked in outpatient and ambulatory

**Table 1. Intention and reasons of physicians in Canada for quitting their current job within two years.**

| Factor | Number* | (%)# |
|---|---|---|
| **Intention to quit current job within 2 years** | | |
| Yes | 19509 | 20.8 |
| No | 74443 | 79.2 |
| **Reasons for quitting job#** | | |
| Stress or burnout or mental health concerns | 7034 | 7.5 |
| Retiring | 10467 | 11.1 |
| Lack of job satisfaction | 4722 | 5.0 |
| Concerns physical health | 2296 | 2.4 |
| Concerns about household members | 1941 | 2.1 |
| Financial impacts | 1076 | 1.1 |
| Health care system | 3443 | 3.7 |
| Other career opportunity | 4299 | 4.6 |
| Other | 2243 | 2.4 |

#Percentages do not add to 100% because of multiple answers.

**Table 2. Characteristics of participants and relationships between socio-demographic, job and health-related factors and quitting due to mental health reasons among physicians in Canada.**

| Factor | %* | OR | 95%CI | p-value | Adjusted OR | 95%CI | p-value |
|---|---|---|---|---|---|---|---|
| **Job-related factors** | | | | | | | |
| **Mode of healthcare delivery** | | | | | | | |
| In-person | 32.9 | 1 | | | 1 | | |
| Online | 7.6 | 1.07 | 0.98, 1.18 | 0.149 | 0.66 | 0.58, 0.75 | **<0.001** |
| Blended | 59.4 | 0.92 | 0.88, 0.97 | **0.002** | 0.67 | 0.63, 0.72 | **<0.001** |
| **Years worked in the job** | | | | | | | |
| Less than 10 years | 35.5 | 0.75 | 0.71, 0.79 | **<0.001** | 1.45 | 1.23, 1.71 | **<0.001** |
| 10 to 19 years | 17.9 | 0.41 | 0.37, 0.44 | **<0.001** | 0.61 | 0.54, 0.70 | **<0.001** |
| 20 years or more | 35.0 | 1 | | | 1 | | |
| Other# | 11.6 | 0.91 | 0.85, 0.99 | 0.020 | 1.17 | 1.07, 1.29 | **0.001** |
| **Number of locations worked at** | | | | | | | |
| One location | 28.3 | 1.22 | 1.16, 1.29 | **<0.001** | 1.19 | 1.12, 1.26 | **<0.001** |
| Multiple locations | 65.6 | 1 | | | 1 | | |
| **Type of healthcare job location** | | | | | | | |
| Acute care | 54.6 | 1 | | | 1 | | |
| Outpatient and ambulatory care | 40.5 | 1.19 | 1.13, 1.25 | **<0.001** | 1.32 | 1.23, 1.41 | **<0.001** |
| Other# | 2.6 | 1.25 | 1.08, 1.45 | **0.003** | 0.92 | 0.76, 1.12 | 0.405 |
| **Received formal training on IPC** | | | | | | | |
| Yes | 78.5 | 0.83 | 0.78, 0.88 | **<0.001** | 1.19 | 1.10, 1.28 | **<0.001** |
| No/ Workplace do not have an IPC policy/valid skip | 20.7 | 1 | | | 1 | | |
| **Socio-demographic factors** | | | | | | | |
| **Gender** | | | | | | | |
| Male | 50.9 | 1 | | | 1 | | |
| Female | 48.9 | 1.26 | 1.20, 1.33 | **<0.001** | 1.81 | 1.71, 1.92 | **<0.001** |
| **Age** | | | | | | | |
| 18 to 34 years | 21.7 | 1 | | | 1 | | |
| 35 to 44 years | 25.5 | 1.07 | 0.99, 1.15 | 0.088 | 3.23 | 2.85, 3.65 | **<0.001** |
| 45 to 54 years | 21.3 | 1.08 | 1.00, 1.17 | 0.058 | 4.62 | 3.87, 5.51 | **<0.001** |
| 55 years and older | 31.5 | 1.66 | 1.55, 1.77 | **<0.001** | 6.58 | 5.44, 7.95 | **<0.001** |
| **Province of residence** | | | | | | | |
| Atlantic provinces Quebec | 5.9 | 1.90 | 1.67, 2.17 | **<0.001** | 3.09 | 2.67, 3.59 | **<0.001** |
| Quebec | 24.2 | 1 | | | 1 | | |
| Ontario | 38.5 | 3.17 | 2.92, 3.43 | **<0.001** | 4.19 | 3.80, 4.62 | **<0.001** |
| Manitoba | 3.4 | 2.48 | 2.14, 2.87 | **<0.001** | 3.43 | 2.89, 4.06 | **<0.001** |
| Saskatchewan | 2.6 | 2.00 | 1.68, 2.38 | **<0.001** | 1.89 | 1.48, 2.41 | **<0.001** |
| Alberta | 11.3 | 3.25 | 2.95, 3.58 | **<0.001** | 3.77 | 3.36, 4.23 | **<0.001** |
| British Columbia | 14.1 | 1.85 | 1.67, 2.04 | **<0.001** | 1.98 | 1.75, 2.24 | **<0.001** |
| **Household income** | | | | | | | |
| ≤$149,999 | 9.4 | 1.45 | 1.35, 1.57 | **<0.001** | 2.71 | 2.43, 3.04 | **<0.001** |
| ≥ 150,000 | 76.2 | 1 | | | 1 | | |
| Not stated | 14.3 | 1.52 | 1.43, 1.62 | **<0.001** | 1.34 | 1.24, 1.44 | **<0.001** |
| **Immigration status** | | | | | | | |
| Non-immigrant | 70.1 | 0.87 | 0.82, 0.92 | **<0.001** | 1.20 | 1.12, 1.28 | **<0.001** |
| Immigrant or non-permanent resident | 23.6 | 1 | | | 1 | | |
| **Health-related factors** | | | | | | | |
| **Perceived general health** | | | | | | | |

*(Continued)*

**Table 2.** (Continued)

| Factor | %* | OR | 95%CI | p-value | Adjusted OR | 95%CI | p-value |
|---|---|---|---|---|---|---|---|
| Poor/Fair | 5.6 | 4.19 | 3.89, 4.51 | <0.001 | 5.39 | 4.95, 5.87 | <0.001 |
| Good | 24.3 | 1.51 | 1.42, 1.59 | <0.001 | 1.68 | 1.58, 1.79 | <0.001 |
| Very good/Excellent/ | 69.9 | 1 | | | 1 | | |

care job settings were more likely to have the intention to leave their profession for mental health concerns than those who worked in acute care (OR = 1.32, 95% CI: 1.23–1.41). Similarly, the intention to quit their job due to stress, burnout or mental health concerns was significantly higher among those who received formal training in Infection Prevention and Control (IPC) than those who did not receive training. Concerning sociodemographic-related factors, females were significantly more likely to intend to leave their profession due to stress, burnout, or mental health concerns (OR = 1.80, 95% CI: 1.70–1.92). Intention to leave the profession was significantly higher among those aged 55 years and older than younger counterparts aged 18–34 (OR = 6.58, 95% CI: 5.44–7.95). Likewise, those participants who resided in Ontario had the highest likelihood of intending to quit their jobs than those who lived in Quebec (OR = 4.19, 95% CI: 3.80–4.6). Furthermore, individuals who reported a household income of ≤$149,999 were 2.7 times at increased odds for intention to quit their job due to mental health reasons than those with a household income of ≥ 150,000. Lastly, regarding health-related factors, perceived general health was significantly associated with the intention to leave the profession. Participants who rated their general health as poor and fair or good were at significantly higher odds of quitting their job than those who perceived their general health as excellent (OR = 5.39, 95% CI: 4.95–5.87; OR = 1.68, 95% CI: 1.58–1.79, respectively).

## Discussion

The COVID-19 pandemic prompted adaptations to healthcare delivery services across Canada. Implementing telemedicine was one strategy for mitigating risk and the spread of infection among patients and medical staff [2]. The present study investigated the impact of modes of healthcare delivery and the intention to quit the workforce due to stress, burnout, or mental health among Canadian physicians during the pandemic. Results showed that 7.5% of the physicians reported an intention to exit the workforce within two years due to stress, burnout, or mental health concerns. Furthermore, virtual or blended modalities of healthcare delivery were associated with decreased odds of intention to quit job (OR = 0.67, 95% CI: 0.63–0.72 and OR = 0.66, 95% CI: 0.58–0.75 respectively) compared to those providing in-person care during the pandemic. The data presented here sheds light on the importance of developing and implementing strategies that promote the well-being of physicians. Incorporating such strategies may mitigate mental health risks among physicians, which in turn can enhance the quality of patient care.

The pandemic subjected physicians to distressing conditions involving fear of contracting the virus, quarantine protocols after exposure (or possible exposure) and social stigma due to being frontline workers [16]. Accordingly, healthcare protocols incorporating flexible patient care modalities have benefited physicians, acting as protective factors against COVID-19-related professional and personal challenges. This may have contributed to differences in retention between those working solely in person and physicians who could incorporate telemedicine into their medical practice. The data gathered in the present paper aligns with past research that has shown telemedicine to decrease the risk of burnout among physicians during

the pandemic. Combined with mitigating burnout, it has also been reported that telemedicine is a cost-effective strategy for both physicians and patients [17]. Furthermore, efficiencies in patient care and increased productivity have also been reported, alongside improved quality of life [18,19]. Regarding a blended modality of care specifically, studies during the pandemic highlighted some concerns physicians reported on challenges surrounding virtual assessments and diagnoses and how inadequacies could negatively impact patient health [6]. Thus, having the ability to provide both in-person and online care may have alleviated those concerns, allowing physicians to perform assessments in person and follow their patients' treatment plans virtually. Providing both in-person and online care during the endemic phase of COVID-19, may offer patients safety and provide continuity of care, especially for those with chronic conditions. In addition, providing both in-person and online care may provide increased flexibility and time efficiency for physicians.

Job-related factors, including years worked on the job and the type of healthcare job setting, were significantly associated with the intention to exit the workforce. Physicians with less than ten years of experience were more likely to report an intention to quit than those who had been in the field for more than 20 years. It is understandable that navigating challenging medical circumstances takes extensive training and practice. Thus, compared to their colleagues who had more years in the medical field, newer physicians may have had less of a foundation to draw on to address increased workloads and job strain. Furthermore, those who worked in outpatient and ambulatory care were more likely to report an intention to leave compared to those working at acute care locations. The data surrounding mental health challenges between outpatient and inpatient (acute care) physicians may outline job-related differences between settings, as substantiated by previous findings from other studies. For instance, a systematic review assessing the literature on burnout among physicians found that those working in an outpatient setting reported higher levels of exhaustion compared to those in an inpatient setting [20]. These outcomes may be attributed to several protective factors facilitating better coping with high job-related stress. For one, inpatient/acute care physicians may have more opportunities for teamwork, which has been found to reduce burnout and other mental health challenges [21]. Physicians in acute settings may also have more opportunities to see the immediate impacts of their care on patients, which can be rewarding. Furthermore, previous studies have found that outpatient physicians report challenges with workloads and administrative tasks among factors associated with burnout [22].

Interestingly, while the strength of the relationship was not substantial, receiving formal training in Infection Prevention Control (IPC) was significantly associated with increased odds of intention to quit job due to stress, burnout, or mental health concerns than not receiving such training. IPC training educates healthcare workers on safe workplace practices that mitigate health-related risks. The need for formal training in IPC may suggest that workplace were high risk and more stressful areas to work in. Although further investigation is needed to explain this finding, it is possible that the results observed relationship is confounded by other variables.

Sociodemographic results showed that females were at greater odds of stress, burnout, and mental health, impacting their intention to quit their jobs compared to males. This is similar to previous research findings from a systematic review, reporting females to be at a greater risk of burnout and emotional exhaustion [23]. Females may feel added pressure to demonstrate high-performance results in their respective fields. Moreover, navigating caregiving responsibilities could have been a contributing factor, as home-related changes during the pandemic reportedly correlated with emotional exhaustion among male and female physicians, with cumulative effects per change [24]. These findings are consistent with data presented in Medscape's National Physician Burnout and Suicide report, which showed that 51% of physicians

who felt burnout were female [25]. Furthermore, compared to non-immigrants, immigrants were less likely to intend to leave their jobs due to stress, burnout, or mental health. Previous studies have found similar findings, noting that being an immigrant was a protective factor against burnout. A possible explanation for this may be a result of life experiences. Given the complex and competitive nature of securing residency and integration-related processes, immigrants may be more resilient to burnout in this context [24,26]. When compared to physicians 18–34 age bracket, those 55 or older were at increased odds for quitting jobs for stress, burnout, and mental health. Out of those who intend to quit current job within 2 years, 11.7% reported both "retiring" and "stress or burnout or mental health concerns" as reasons for their intention to quit. This may explain why physicians 55 or older were more likely to have this intention. Earnings also appeared to predict the same risks as those earning less than $149,999 were at a greater risk than those earning more than $150,000. It may be the case that less experience on the job correlates with a lower income compared to colleagues with more experience. These findings are important for future considerations regarding appropriate compensation for frontline workers tasked with assessing and treating patients during a public health crisis while managing individual risks for contracting infection.

Our results highlighted that physicians 55 or older were at an increased odds for the same reasons. Global public health data underscored the associated risks of COVID-19 as being higher for older adults [27]. As frontline workers, physicians were in a difficult position as they could not always maintain physical distancing protocols. For older physicians, these risks may have contributed to fear of contracting COVID-19, negatively impacting mental health. This likely ties into other data obtained from this study, which showed that those who reported poor/fair health were at an increased risk of intention to exit the workforce due to stress, burnout, or mental health compared to those who reported good health. Among at-risk populations, which included older adults, were those with serious health conditions such as diabetes, obesity, and cardiovascular disease [28].

This study was unique in that it was the first to examine the relationship between the mode of healthcare delivery and physicians' intention to quit their jobs within two years due to stress, burnout, and mental health during the COVID-19 pandemic. Population weights were used to accurately represent the Canadian population of physicians. It is important to note, however, that this study has a few limitations. Given this study's cross-sectional nature, we are unable to infer causation and data relied on self-report measures, possibly subjecting the study to information bias. In addition, there is a possibility of selection bias as response rate was 54.9% and the data was collected end of 2021, 2 years after the pandemic started. It is possible that physicians who really wanted to quit have already done so earlier before the survey was done. Furthermore, the study did not assess factors such as physicians' different specialties or subspecialties, resilience or personality, which could have been confounding variables in this study.

Physicians are an integral part of medical practice, and their retention is unequivocally linked to the quality of patient care. Therefore, incorporating a framework that supports physicians' mental health and job-related needs is critical for enhancing healthcare infrastructure. These components are a likely precursor to the level of job satisfaction that physicians will feel over time in their medical practice. As the COVID-19 pandemic contributed to medical workforce adaptations, it is important to consider how changes in clinical approach factored into mental health and the desire to remain a working physician. This understanding is important in highlighting the gap in current knowledge and the resources that might contribute to physicians' capacity to stay on the job. This is especially critical under emergency circumstances where, as with COVID-19, traditional practices and procedures are challenging or impossible to execute. This understanding could further be incorporated into training and education

programs to enhance future preparedness. Thus, underscoring contributing factors related to retention or exiting the workforce can help healthcare systems incorporate implementation strategies that support and benefit physicians, thereby optimizing patient care.

## Author Contributions

**Conceptualization:** Hossam Ali-Hassan.

**Formal analysis:** Safoura Zangiabadi.

**Supervision:** Hossam Ali-Hassan.

**Writing – original draft:** Hossam Ali-Hassan, Shauna Clayton, Safoura Zangiabadi.

**Writing – review & editing:** Hossam Ali-Hassan, Shauna Clayton.

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
