## [Decision Letter · Decision Letter 0]

30 Apr 2024

PGPH-D-24-00506

Effect of mode of healthcare delivery on stress and intention to quit among physicians in Canada during the COVID-19 pandemic

Dear Dr. Ali-Hassan,

Thank you for submitting your manuscript to PLOS Global Public Health. After careful consideration, we feel that it has merit but does not fully meet PLOS Global Public Health’s publication criteria as it currently stands. Therefore, we invite you to submit a revised version of the manuscript that addresses the points raised during the review process.

We look forward to receiving your revised manuscript.

Kind regards,

Sok King Ong

Academic Editor

Journal Requirements:

Additional Editor Comments (if provided):

Reviewers' comments:

Reviewer's Responses to Questions

**Comments to the Author**

1. Does this manuscript meet PLOS Global Public Health’s publication criteria? Is the manuscript technically sound, and do the data support the conclusions? The manuscript must describe methodologically and ethically rigorous research with conclusions that are appropriately drawn based on the data presented.

Reviewer #1: Partly

Reviewer #2: Yes

2. Has the statistical analysis been performed appropriately and rigorously?

Reviewer #1: Yes

Reviewer #2: Yes

3. Have the authors made all data underlying the findings in their manuscript fully available (please refer to the Data Availability Statement at the start of the manuscript PDF file)?

Reviewer #1: Yes

Reviewer #2: Yes

4. Is the manuscript presented in an intelligible fashion and written in standard English?

Reviewer #1: No

Reviewer #2: Yes

5. Review Comments to the Author

Reviewer #1: The paper describes results from a cross-sectional study looking at healthcare workers experiences during the pandemic.

1. As the authors mentioned in the limitation section, the cross-sectional nature of the study meant it was not possible to infer causation. Some of the remarks may need to be more cautious; e.g. in the abstract, it states online or blended healthcare delivery decreased the odds of intention to quit, while it may be more accurate to say it was associated with a decreased odds.

2. In the last paragraph of the Introduction, the Poland study showed that high levels of stress and fear was associated with job satisfaction - presumably this was 'negatively' associated

3. Response rate was 54.9% and data was collected end of 2021 (2 years after the pandemic started). It is unclear whether physicians who really wanted to quit have already done so earlier before the survey was done, and whether response bias could have played a role - those who wanted to quit were less likely to complete the survey. This may need to be clarified or included as a limitation.

4. In the results section, 20.8% expressed intention to quit their job, while 7.5% were due to stress, burnout or mental health concerns. According to the table, 11.1% were retiring anyway, which explains half of the people intending to quit. This may also explain why more physicians age >55 years were more likely to have this intention, or perhaps not so interested to adjust to online / blended healthcare delivery. What do the author's think about this?

5. The authors in the discussion section stated that providing both in-person and online care may reduce concerns from physicians re: providing care to patients. What are the implications to practice now in the endemic phase of COVID-19?

6. IPC formally trained were likely to quit their job. The authors suggested that this may be due to pressure to adhere to safety protocols. In my opinion, the physicians would want to adhere to the safety protocols given IPC measures are risk mitigation measures to reduce infection risk. However, the need for formal training in IPC suggests that these were high risk and more stressful areas to work in, so there may be other confounding factors for this finding.

There are a few areas to clarify, especially in the discussion section. I look forward to hearing the feedback and responses from the authors to clarify these issues.

Reviewer #2: Well written manuscript on the "Effect of mode of healthcare delivery on stress and intention to quit among physicians

in Canada during the COVID-19 pandemic".

Did the survey look at the specialty of the responders? Studies have shown that physicians in different specialties or sub-specialties encounter different levels of stress during the pandemic. For example, a emergency surgeon's level of stress would be expected to be higher than a psychiatrist.

The survey was conducted in 2021, was this during the height of the pandemic restrictions in Canada? Does this correspond to the movement restrictions or period of lock down?

6. PLOS authors have the option to publish the peer review history of their article (what does this mean?). If published, this will include your full peer review and any attached files.

**Do you want your identity to be public for this peer review?** For information about this choice, including consent withdrawal, please see our Privacy Policy.

Reviewer #1: **Yes: **Shyh Poh Teo

Reviewer #2: No

---

## [Decision Letter · Decision Letter 1]

21 Jun 2024

Effect of mode of healthcare delivery on stress and intention to quit among physicians in Canada during the COVID-19 pandemic

PGPH-D-24-00506R1

Dear Dr. Ali-Hassan,

We are pleased to inform you that your manuscript 'Effect of mode of healthcare delivery on stress and intention to quit among physicians in Canada during the COVID-19 pandemic' has been provisionally accepted for publication in PLOS Global Public Health.

Best regards,

Sok King Ong

Academic Editor

Reviewer Comments (if any, and for reference):

Reviewer's Responses to Questions

**Comments to the Author**

1. If the authors have adequately addressed your comments raised in a previous round of review and you feel that this manuscript is now acceptable for publication, you may indicate that here to bypass the “Comments to the Author” section, enter your conflict of interest statement in the “Confidential to Editor” section, and submit your "Accept" recommendation.

Reviewer #1: All comments have been addressed

Reviewer #2: All comments have been addressed

2. Does this manuscript meet PLOS Global Public Health’s publication criteria? Is the manuscript technically sound, and do the data support the conclusions? The manuscript must describe methodologically and ethically rigorous research with conclusions that are appropriately drawn based on the data presented.

Reviewer #1: Yes

Reviewer #2: Yes

3. Has the statistical analysis been performed appropriately and rigorously?

Reviewer #1: Yes

Reviewer #2: Yes

4. Have the authors made all data underlying the findings in their manuscript fully available (please refer to the Data Availability Statement at the start of the manuscript PDF file)?

Reviewer #1: Yes

Reviewer #2: Yes

5. Is the manuscript presented in an intelligible fashion and written in standard English?

Reviewer #1: Yes

Reviewer #2: Yes

6. Review Comments to the Author

Reviewer #1: All the comments made were addressed by the authors.

Reviewer #2: Thank you very much for responding to my comments and submitting the revised manuscript.

7. PLOS authors have the option to publish the peer review history of their article (what does this mean?). If published, this will include your full peer review and any attached files.

**Do you want your identity to be public for this peer review?** For information about this choice, including consent withdrawal, please see our Privacy Policy.

Reviewer #1: **Yes: **Shyh Poh Teo

Reviewer #2: **Yes: **Kenneth Y Y Kok
